# Riboflavin 1 Transporter Deficiency: Novel *SLC52A1* Variants and Expansion of the Phenotypic Spectrum

**DOI:** 10.3390/genes14071408

**Published:** 2023-07-07

**Authors:** Sarah C. Grünert, Athanasia Ziagaki, André Heinen, Anke Schumann, Sara Tucci, Ute Spiekerkoetter, Miriam Schmidts

**Affiliations:** 1Department of General Pediatrics, Adolescent Medicine and Neonatology, Medical Center, Faculty of Medicine, University of Freiburg, 79106 Freiburg, Germany; anke.schumann@uniklinik-freiburg.de (A.S.); sara.tucci@uniklinik-freiburg.de (S.T.); ute.spiekerkoetter@uniklinik-freiburg.de (U.S.); miriam.schmidts@uniklinik-freiburg.de (M.S.); 2Medizinische Klinik für Endokrinologie und Stoffwechselmedizin, Campus Virchow, Charité Universitätsmedizin Berlin, 10117 Berlin, Germany; athanasia.ziagaki@charite.de; 3Department of Pediatrics, Faculty of Medicine and University Hospital Carl Gustav Carus, Technische Universität Dresden, 01307 Dresden, Germany; andre.heinen@ukdd.de; 4Pharmacy, Medical Center, University of Freiburg, 79106 Freiburg, Germany; 5CIBSS—Center for Integrative Biological Signaling Studies, University of Freiburg, 79104 Freiburg, Germany

**Keywords:** riboflavin, riboflavin transporter, MADD, *SLC52A1*, vitamin B2

## Abstract

Riboflavin transporter 1 (RFVT1) deficiency is an ultrarare metabolic disorder due to autosomal dominant pathogenic variants in *SLC52A1*. The RFVT1 protein is mainly expressed in the placenta and intestine. To our knowledge, only five cases of RFVT1 deficiency from three families have been reported so far. While newborns and infants with *SLC52A1* variants mainly showed a multiple acyl-CoA dehydrogenase deficiency-like presentation, individuals identified in adulthood were usually clinically asymptomatic. We report two patients with novel heterozygous *SLC52A1* variants. Patient 1 presented at the age of 62 with mild hyperammonemia following gastroenteritis. An acylcarnitine analysis in dried blood spots was abnormal with a multiple acyl-CoA dehydrogenase deficiency-like pattern, and genetic analysis confirmed a heterozygous *SLC52A1* variant, c.68C > A, p. Ser23Tyr. Patient 2 presented with recurrent seizures and hypsarrhythmia at the age of 7 months. Metabolic investigations yielded unremarkable results. However, whole exome sequencing revealed a heterozygous start loss variant, c.3G > A, p. Met1Ile in *SLC52A1.* These two cases expand the clinical spectrum of riboflavin transporter 1 deficiency and demonstrate that symptomatic presentation in adulthood is possible.

## 1. Introduction

Riboflavin, also called vitamin B2, is a water-soluble essential vitamin that is present in dietary sources such as meats, milk, fatty fish, nuts and eggs as well as certain vegetables and fruits [1]. Riboflavin is a precursor in the synthesis of flavin mononucleotide (FMN) and flavin adenine dinucleotide (FAD). The derivatives FMN and FAD play crucial roles as obligate cofactors of numerous enzymes called flavoproteins [1,2]. Many of these flavoproteins are located in the mitochondria, among them are electron transfer flavoprotein (ETF) and ETF ubiquinone oxidoreductase (ETF:QO). ETF acts as an intermediary electron carrier for at least 12 flavoprotein dehydrogenases, including 10 acyl-CoA dehydrogenases involved in fatty acid oxidation or amino acid metabolism, as well as two flavoprotein dehydrogenases involved in choline metabolism [2]. ETF:QO is needed for the electron transfer to ubiquinone in the respiratory chain.

Multiple acyl-CoA dehydrogenase deficiency (MADD), or glutaric aciduria type II, is an autosomal recessive disorder caused by biallelic mutations in the genes encoding for ETF or ETF:QO [3]. As a defect of fatty acid and amino acid metabolism it is characterized by a risk of metabolic decompensations with metabolic acidosis and hyperammonemia. Late onset forms mainly present with muscular symptoms and lipid droplet myopathy [3]. MADD has a pathognomonic metabolite pattern; urinary organic acid analysis usually shows elevated levels of glutaric, ethylmalonic, 3-hydroxyisovaleric, 2-hydroxyglutaric, 5-hydroxyhexanoic, adipic, suberic, sebacic and dodecanedioic acid without relevant ketonuria, as well as glycine conjugates of C4 and C5 acids [3,4]. The acylcarnitine profile typically shows increased concentrations of several short-, medium- and long-chain acylcarnitines, such as C4, C5, C5-DC, C6, C8, C10, C12, C14:1, C16 and C18:1 [3,4].

An MADD-like metabolite pattern has also been reported in newborns of mothers with riboflavin deficiencies [2]. In addition, within the last decade, different genetic defects of riboflavin metabolism (including intercellular riboflavin transport, FAD biosynthesis and FAD transport) have been shown to be linked with MADD-like phenotypes [5].

The absorption and transportation of riboflavin is performed by a group of transporters from the solute carrier family SLC52 [1]. The majority of riboflavin is absorbed in the small intestine, but absorption also occurs in the stomach, duodenum, colon and rectum by active transport through the riboflavin transporter 3 (RFVT3) [1]. Within the epithelial cells of the gastrointestinal tract, riboflavin can either be transformed into FMN and FAD or transported to the bloodstream by riboflavin transporter 1 (RFVT1) or riboflavin transporter 2 (RFVT2) for further distribution to other tissues. 

RFVT1 was the first riboflavin transporter detected. Its identification and functional characterization was reported by Yonezawa et al. in 2008 [6]. *SLC52A1* encoding the RFVT1 protein is mainly expressed in the placenta and intestine [7]. It therefore plays an important role for the placental transport of maternal riboflavin to the fetus [8]. RFVT1 deficiency is caused by autosomal dominant, heterozygous *SLC52A1* pathogenic variants. To our knowledge, only five cases of RFVT1 deficiency from three families have been reported so far [2,8,9,10]. While newborns and infants with *SLC52A1* variants mainly showed MADD-like symptoms, individuals identified in adulthood were usually asymptomatic apart from a typical biochemical profile. We herein report two patients with novel heterozygous *SLC52A1* variants.

## 2. Materials and Methods

### 2.1. Genetic Testing

Genetic testing was performed as part of routine clinical care to obtain a clinical diagnosis after appropriate consent of the patients or their legal guardians. DNA was isolated from EDTA blood samples, and in patient 1, a genetic panel analysis for fatty acid oxidation defects was performed, while in patient 2, a genetic whole exome analysis was performed.

### 2.2. Protein Modelling

*SLC52A1* protein structures were predicted using Phyre 2 (http://www.sbg.bio.ic.ac.uk/phyre2/html/page.cgi?id=index (accessed on 19 March 2023) and the effect of patient alleles was predicted using Missense3D (http://missense3d.bc.ic.ac.uk/ (accessed on 19 March 2023).

## 3. Results

### 3.1. Case Presentations

#### 3.1.1. Case 1

The patient is a 64-year-old German woman. She presented at the age of 62 with a first episode of mild hyperammonemia (74 µmol/L) following gastroenteritis. This episode was clinically characterized by reduced vigilance, nausea and vomiting. Under treatment with ornithine, aspartate and lactulose, the ammonia concentration normalized. The patient suffered from general fatigue, chronic headaches and joint pain, as well as episodic chest discomfort. Previous diagnoses included coronary artery disease, chronic obstructive pulmonary disease, dyslipidemia and severe depression. At presentation to the metabolic clinic, she was on ASS 100 mg/day, simvastatin 20 mg/day, metoprolol 47. 5 mg/day and ranolazin 375 mg twice daily. She had no known allergies, drank no alcohol, but was a smoker at 20 packs years. She used to be physically active and liked to cycle. She has not been able to work since the age of 53 due to her psychiatric diagnosis. She had a 37-year-old healthy son. The family history revealed that the mother of the patient had also been suffering from severe depression. 

Clinical examination was unremarkable except for a systolic murmur. Her BMI was 22.4 kg/m², and she was cognitively normal. An amino acid analysis in plasma as well as urine organic acids yielded normal results. Due to the mild hyperammonemia, Sanger sequencing of the *OTC* gene was performed to exclude heterozygous ornithine transcarbamylase deficiency (OTC deficiency), but no pathogenic variant was detected. An acylcarnitine analysis in dried blood spots, however, was abnormal with mild elevation of the medium-chain acylcarnitines C8, C10 and C10:1. Concentrations of C14:2 and C16:2, as well as the acylcarnitine ratios C14:1/C16 and C8/C2 were also increased. This metabolite pattern was suggestive of either multiple acyl-CoA dehydrogenase deficiency (MADD) or disorders of the riboflavin metabolism. A genetic panel analysis was performed and yielded a heterozygous variant, c.68C > A (p. Ser23Tyr), in the *SLC52A1* (NM_017986.4) gene, confirming the diagnosis of an RFVT1 deficiency. At the time of diagnosis, the patient was hospitalized due to a new major depressive episode. Riboflavin treatment in a dosage of 100 mg daily was initiated. Due to the patient’s hospitalization, no further investigations were accomplished.

#### 3.1.2. Case 2

The girl is the first child of non-consanguineous German parents. Pregnancy, birth and early infancy were unremarkable. At the age of 7 months, the child presented with recurrent, daily seizures. At that time, she showed normal growth and development. The EEG revealed generalized hypsarrhythmia, and therapy with pyridoxalphosphate and folic acid was initiated. Additionally, she received three pulses of dexamethasone. Since the first dexamethasone pulse, no further seizures occurred, and the EEG improved significantly. A brain MRI was unremarkable. Microbiologic, virologic and metabolic investigations yielded normal results. In particular, the acylcarnitine analysis in dried blood spots as well as the organic acid analysis in urine were unremarkable. Riboflavin concentration was not measured. The further clinical course was uneventful. Developmental testing at the age of 19 months (Bailey III) showed normal cognitive and speech development, but mildly impaired motor development. Whole exome sequencing was performed and yielded the heterozygous start loss variant, c.3G > A, p. Met1Ile in *SLC52A1* (NM_017986.4). The variant was classified as a variant of unknown significance. The same variant was detected in her mother. Until the age of 2.75 years, no further symptoms occurred.

## 4. Discussion

While RVFT2 and RVFT3 deficiencies are clinically well-characterized disorders, only five cases of RFVT1 deficiency from three families have been reported so far [2,8,9,10]. An overview of the genetic findings as well as clinical symptoms of all known cases is shown in Table 1. It is of interest that so far only newborns and infants were reported to present with clinical symptoms mimicking MADD, while their parents identified in adulthood were usually asymptomatic. Patient 1 in this study is the only patient with mild symptoms that presented in adulthood. 

Ho et al. [9] reported a woman with riboflavin deficiency in whom a heterozygous de novo 1.9-kb deletion in the *SLC52A1* gene was identified. The woman was clinically asymptomatic, but showed biochemical evidence of a riboflavin deficiency in terms of increased serum acylcarnitines. She was originally diagnosed after her newborn daughter presented with poor sucking, hypoglycemia and metabolic acidosis on the first day of life [2]. The child had dicarboxylic aciduria and elevated plasma acylcarnitine levels, initially thought to be consistent with MADD. Oral riboflavin supplementation resulted in complete resolution of the clinical and biochemical findings. The findings were consistent with transient neonatal riboflavin deficiency secondary to maternal riboflavin deficiency that was exacerbated during pregnancy. The infant did not carry the deletion identified in the mother.

Mosegaard et al. reported a very similar case of a child with a transient neonatal riboflavin deficiency [8]. The acylcarnitine profile was again suggestive of MADD. Genetic analyses, however, identified heterozygosity for an intronic variant (c.234 + 11G > A) in *SLC52A1*. Heterozygosity for the same variant was confirmed in the mother who showed neither clinical nor biochemical symptoms apart from a borderline low riboflavin level.

These case reports show that RFVT 1 is essential for placental transport of riboflavin, and that impaired riboflavin transport due to RFVT1 haploinsufficiency can result in an MADD-like clinical and biochemical phenotype of the newborn, even if the newborn himself does not carry the *SLC52A1* variant. The metabolic decompensation may be life-threatening with encephalopathy and hyperammonemia [8], and neither clinical symptoms nor biochemical findings allow distinguishing RFVT1 deficiency from MADD deficiency (caused by biallelic *ETFA, ETFB* and *ETFDH* variants). Metabolic decompensation can, however, be easily corrected and prevented by riboflavin supplementation. Both mothers suffered from severe hyperemesis gravidarum, suggesting that catabolism in pregnancy may be an important risk factor [1].

Kang et al. [10] reported a previously healthy 4-month-old girl that presented with recurrent seizures. There were no prenatal or perinatal problems, and she showed normal growth and development. A blood gas analysis at admission revealed compensated metabolic acidosis (pH 7.275, PCO2 36.3 mm Hg, bicarbonate 17.0 mmol/L, base excess –8.4) and hyperammonemia of 170 μmol/L. The brain MRI was normal, and EEG showed interictal epileptiform discharges over the midline central region. Hyperammonemia was initially considered to be caused by the seizure, but after a transient decrease in ammonia levels under dextrose treatment, ammonia concentrations increased again to a maximum of 208 µmol/L after discontinuation of dextrose infusions. Genetic diagnostics revealed a homozygous exon 3 deletion in *SLC52A1* and heterozygous deletions in exons 1, 2, 4 and 5. Testing of her parents revealed heterozygous deletions in exons 1–5 of *SLC52A1* in her father and a 25% decrease in DNA concentration in the mother, indicating a 25% allele frequency for the exon 3 deletion allele (mosaic heterozygous exon 3 deletion) in the child. Her father was healthy and had no symptoms related to this mutation; however, no biochemical analyses are reported [10]. 

Patient 1 of our study is the oldest patient with an RFVT1 deficiency reported so far. It is of note that she presented with mild hyperammonemia triggered by a gastrointestinal infection. Similar to the patient reported by Chiong et al. [2], she showed a mild biochemical MADD pattern even during times of well-being. Unfortunately, her riboflavin level before the start of treatment is not available. In contrast, patient 2 had a normal acylcarnitine profile. It is difficult to assess whether the clinical symptoms of our patients can really be attributed to RFVT1 deficiency, but the clinical presentation of patient 2 resembles the clinical phenotype of the patient reported by Kang et al. [10]. However, the genetic variant identified in patient 2 is classified as a variant of unknown significance. Since the same variant was found in the mother, an MADD-like neonatal presentation of the child could have been expected as reported by Mosegaard et al. [8].

Figure 1 shows an overview of *SLC25A1* disease variants. While all previously reported patients carried either large deletions or *SLC52A1* splice site variants that are expected to be associated with a loss of function of the RFVT1 protein, our patients were heterozygous for two novel variants representing a start-loss and a missense variant. Both variants are predicted to be pathogenic using an in silico analysis with Alamut Visual 2 software (http://www.interactive-biosoftware.com/alamut-visual/ (accessed on 19 March 2023) that summarizes the results from four different tools covering different functional aspects (SIFT, https://sift.bii.a-star.edu.sg/ (accessed on 19 March 2023); Mutation Taster, http://www.mutationtaster.org/ (accessed on 19 March 2023); PolyPhen-2, http://genetics.bwh.harvard.edu/pph2/ (accessed on 19 March 2023) and Align GVGD, http://agvgd.hci.utah.edu/ (accessed on 19 March 2023)).

The protein modeling including putative effects of the two identified variants is shown in Figure 2. While c.68C > A, p. Ser23Tyr is a missense variant, the c.3G > A, p. Met1Ile allele identified in case 2 represents a start-loss variant. The next in-frame ATG is located at protein position 20, suggesting a protein lacking the first 19 amino acids could result from the variant. The deletion is predicted to have a modest effect on the protein 3D structure; however, loss of a protein localization signal within the deleted first 19 amino acids could result in abrogation of proper protein targeting to the plasma membrane, independent of loss of function effects resulting from changes in the protein structure. Hence, it is possible that this allele represents a true Null allele.

## 5. Conclusions

Our two patients further expand the clinical spectrum of RFVT1 deficiency, demonstrating that onset of symptoms in adulthood is possible. Due to the small number of published cases and clinical heterogeneity, the effect of heterozygous *SLC25A1* gene variants is still not conclusively clear and clinical expression may be influenced by further genetic and extrinsic factors.

## Figures and Tables

**Figure 1 genes-14-01408-f001:**
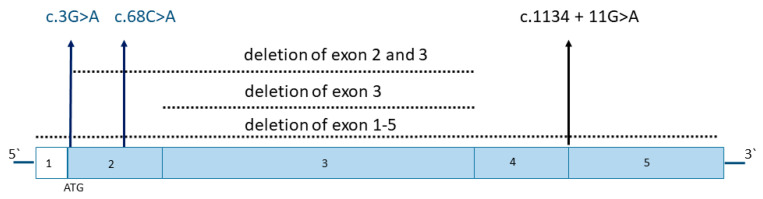
**Overview of *SLC25A1* disease variants.** While all previously reported patients carried either large deletions or *SLC52A1* splice site variants, patients 1 and 2 in this study were heterozygous for novel variants representing a start-loss (c.3G > A) and a missense variant (c.68C > A), respectively. The localization of missense and splice site variants is marked with arrows, deletions are shown as dotted lines indicating the deleted exons.

**Figure 2 genes-14-01408-f002:**
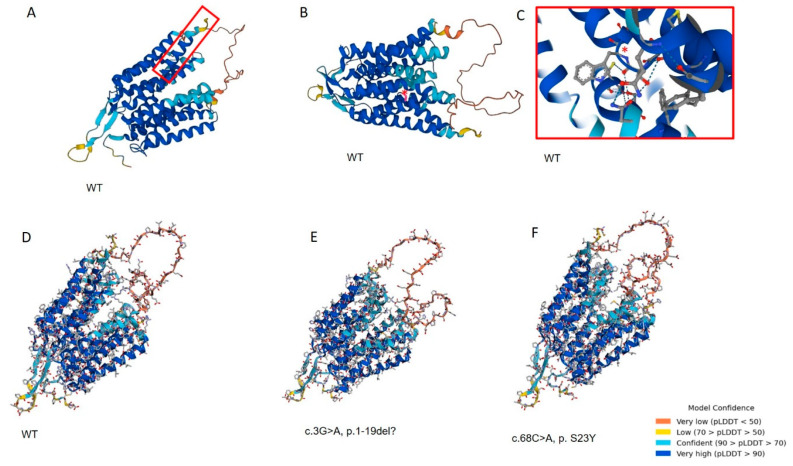
***SLC25A1* 3D structural modelling of wildtype and mutant RFVT1 proteins.** (**A**) Predicted 3D structure of wildtype *SLC52A1*. Amino acids predicted to be deleted in the start-loss alleles are marked with a red square. (**B**) Localization of the p.S23Y variant within the structure is marked in red. (**C**) The red square shows a close-up of the localization of the p.S23Y variant marked with a red star. Structure prediction was performed using Phyre2 (http://www.sbg.bio.ic.ac.uk/phyre2/html/page.cgi?id=index (accessed 19 March 2023). (**D**) Wildtype protein structure compared to *SLC52A1* p. del1-19 (**E**) and p. S23Y (**F**) as predicted by Missense3D (http://missense3d.bc.ic.ac.uk/missense3d/ (accessed on 19 March 2023). While the overall protein 3D structure is maintained for both variants, variants result in a modest modification of the pore shape and size.

**Table 1 genes-14-01408-t001:** Patients with *SLC52A1* variants.

Patient	Genetic Variant	Symptoms	References
#1	heterozygous variant c.68C > A, p. Ser23Tyr	episode of mild hyperammonemia following gastroenteritis	this report
#2	heterozygous variant c.3G > A, p. Met1Ile	infantile seizures (hypsarrhythmia) at 7 months, mild motor delay	this report
#3	heterozygous *de novo* deletion spanning exons 2 and 3	mild MADD profile, but no clinical symptoms, transient MADD-like clinical and biochemical picture in newborn child of this woman (genetic variant not present in the child)	Chiong et al. [2], Ho et al. [9]
#4 (mother of patient #5)	heterozygous variant c.1134 + 11G > A, causing exon 4 skipping	neither clinical nor biochemical symptoms, borderline low riboflavin levels after parturition	Mosegaard et al. [8]
#5 (child of patient #4)	heterozygous variant c.1134 + 11G > A, causing exon 4 skipping	MADD-like clinical and biochemical picture in the newborn child, corrected with riboflavin supplementation	Mosegaard et al. [8]
#6 (child of #7)	homozygous exon 3 deletion and heterozygous deletions in exons 1, 2, 4, and 5	recurrent seizures at 4 months, normal development under riboflavin treatment	Kang et al. [10]
#7 (father of #6)	heterozygous deletions in exons 1–5	no clinical symptoms	Kang et al. [10]

## Data Availability

Data sharing not applicable.

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
