# Peer review of "Riboflavin 1 Transporter Deficiency: Novel SLC52A1 Variants and Expansion of the Phenotypic Spectrum"

_genes, 2023, doi:10.3390/genes14071408_

Round 1

Reviewer 1 Report

The subject of the manuscript is of interest taking into account the few cases reported of T riboflavin transporter 1 deficiency and demonstrated that symptomatic presentation in adulthood could be possible. Although two cases are reported here, the authors make a wide comparison with those reported in the literature.

I think the manuscript fulfils the requirements for publication in the present form.

I suggest minor format changes to a better clarity:

- Page 3, Case 2, lines 148-140: I think there is a mistake when the authors mentiones the age: "Until the age of 2 9/12 years......".

- Page 4, Discusion, lines 164-165: Please put the reference number near the author: "Ho et al. [9]...." instead at the end of the phrase: "Ho et al. reported a woman ........ was identified [9]". The same comment for Page 5, Discusion, lines 190-191: "Kang et al. [10].

- Page 5, Discusion, 1st and 2nd paragraph: please put both paragraph in only one.

Other inquire: I do not understand why the authors said: "Funding: This research received no external funding" and then in the Acknowledgments, they mentioned: "MS received funding......."

Author Response

We thank the reviewer for the very positive evaluation of our article.

This is how we have addressed the reviewer's suggestions:

Page 3, Case 2, lines 148-140: I think there is a mistake when the authors mentiones the age: "Until the age of 2 9/12 years......".

We meant 2 years and 9 months. As this seems to be unclear, we have changed it to 2.75 years.

Page 4, Discusion, lines 164-165: Please put the reference number near the author: "Ho et al. [9]...." instead at the end of the phrase: "Ho et al. reported a woman ........ was identified [9]". The same comment for Page 5, Discusion, lines 190-191: "Kang et al. [10].

We have changed that according to the reviewer's suggestion.

Page 5, Discusion, 1st and 2nd paragraph: please put both paragraph in only one.

We are not quite sure which 2 paragraphs are meant here, because there is the table between the first 2 paragraphs. We would like to leave that up to the editorial office to decide about the layout.

Other inquire: I do not understand why the authors said: "Funding: This research received no external funding" and then in the Acknowledgments, they mentioned: "MS received funding......."

It is correct that no funding was received for this article. The information in the acknowledgement does not refer to this article but to her general funding that should usually be mentioned.

Reviewer 2 Report

enclose file

Author Response

General comments:

  • Genes and de novo in cursive
  • It is better use de word variant that mutation

We have adapted the text according to the reviewer’s suggestions.

  1. Genetic testing. This section could be improved: little information is provided about genetic testing

We thank the reviewer for this comment. However, as genetic testing was done by routine diagnostic methods, we think that additional detailed information is not really interesting for the readers.

  1. Results:
  • The variant c.68C>A describe de protein as is described in case 2
    • We have added information on the protein change in the case report section and the table.
  • There is not specified if variants are pathogenic, likely pathogenic, VUS…
    • We have added the following sentence to the case report section of case 2: “This variant was classified as a variant of unknown significance.”
  • The variant in case 2 is predicted likely pathogenic and variant in case 1 VUS
    • Please see above. It is the other way round.
  • Table 1: describe the variant in same way: c. and p. case 3, 6 and 7 remove the gene (all variants are in SLC52A1)_ de novo deletion exons 2 and 3. Case 4 and 5 remove causing.
    • We have adapted the table according to the reviewer’s suggestions.
  1. Discussion:

Line 177 de variant c.234+11G-A is c.234+11G>A

               We have corrected this.

Line 201: “father and a 25% decrease in the concentration of DNA of exon 3 for the mother”: this sentence is not correct the indication of a mosaicism is done from data of NGS concerning VAF (frequency allelic) not for concentration of DNA

We are really sorry for this mistake. We have changed the sentence as follows: "Testing of her parents revealed heterozygous deletions in exons 1–5 of SLC52A1 in her fa­ther and a 25% decrease of DNA concentration  in the mother, indicating a 25% allele frequency for the exon 3 deletion allele (mosaic heterozygous exon 3 deletion) in the child. "            

Are you sure that variant in case 2 is classified as VUS instead LP?PM2,PVS1,PP5, Null variant in this gene where loss of function is a known mechanism of disease. Explain better

This is already discussed in detail:

In contrast, patient 2 had a normal acylcarnitine profile. It is difficult to assess whether the clinical symptoms of our patients can really be attributed to RFVT1 deficiency, but the clinical presentation of patient 2 resembles the clinical phenotype of the patient reported by Kang et al [10]. However, the genetic variant identified in patient 2 is classified as a variant of unknown significance. Since the same variant was found in the mother, an MADD-like neonatal presentation of the child could have been expected as reported by Mosegaard et al. [8]

AND

While c.68C>A, p.Ser23Tyr is a missense variant, the c.3G>A, p.Met1Ile allele identified in case 2 represents a start loss variant. The next in-frame ATG locates to protein position 20, suggesting a protein lacking the first 19 amino acids could result from the variant. The deletion is predicted to have a modest effect on the protein 3D structure, however, loss of a protein localisation signal within the deleted first 19 amino acids could result in abrogation of proper protein targeting to the plasma membrane, independently of loss of function effects resulting from changes of the protein structure. Hence, it is possible that this allele represents a true Null allele.

Reviewer 3 Report

The authors reported two new patients with variations in the Riboflavin 1 transporter (SLC52A2). These two cases contribute to the expansion of the clinical spectrum associated with riboflavin transporter 1 deficiency. The case report is very straightforward, and the authors provide ample background knowledge about how riboflavin participates in metabolism in mammals. Additionally, the authors compare their new cases with previously reported SLC52A2 mutations in humans, offering a comprehensive understanding of the clinical spectrum associated with this gene mutation.

I have a few suggestions:

Table 1 can be improved. For example, you can provide more details in the "Genetic variant" section, such as specifying that your two cases involve heterozygous mutations. Although you mention this in the context, emphasizing it in the table will make it clearer.

The legend of Figure 1 lacks detail. Please consider improving it by providing more information.

Thank you for your work.

Author Response

The authors reported two new patients with variations in the Riboflavin 1 transporter (SLC52A2). These two cases contribute to the expansion of the clinical spectrum associated with riboflavin transporter 1 deficiency. The case report is very straightforward, and the authors provide ample background knowledge about how riboflavin participates in metabolism in mammals. Additionally, the authors compare their new cases with previously reported SLC52A2 mutations in humans, offering a comprehensive understanding of the clinical spectrum associated with this gene mutation.

We thank the reviewer for this very positive assessment of our article.

I have a few suggestions:

Table 1 can be improved. For example, you can provide more details in the "Genetic variant" section, such as specifying that your two cases involve heterozygous mutations. Although you mention this in the context, emphasizing it in the table will make it clearer.

According to the reviewers suggestion, we have added information on heterozygosity and homozygosity to the table.

The legend of Figure 1 lacks detail. Please consider improving it by providing more information.

We have added the following sentence to provide more information: "The localization of missense and splice site variants is marked with arrows, deletions are shown as dotted lines indicating the deleted exons."